# Process and Time

**DOI:** 10.3390/e25050803

**Published:** 2023-05-15

**Authors:** William Sulis

**Affiliations:** Collective Intelligence Laboratory, McMaster University, 255 Townline Rd. E., Cayuga, ON N0A 1E0, Canada; sulisw@mcmaster.ca; Tel.: +1-905-772-7218

**Keywords:** time, duration, object, process, process algebra, being, becoming, generativity, transience, contextuality, locality, combinatorial games, non-Kolmogorov probability

## Abstract

In regards to the nature of time, it has become commonplace to hear physicists state that time does not exist and that the perception of time passing and of events occurring in time is an illusion. In this paper, I argue that physics is actually agnostic on the question of the nature of time. The standard arguments against its existence all suffer from implicit biases and hidden assumptions, rendering many of them circular in nature. An alternative viewpoint to that of Newtonian materialism is the process view of Whitehead. I will show that the process perspective supports the reality of becoming, of happening, and of change. At the fundamental level, time is an expression of the action of process generating the elements of reality. Metrical space–time is an emergent aspect of relations between process-generated entities. Such a view is compatible with existing physics. The situation of time in physics is reminiscent of that of the continuum hypothesis in mathematical logic. It may be an independent assumption, not provable within physics proper (though it may someday be amenable to experimental exploration).

## 1. Introduction

Einstein, in providing the world with the special and general theories of relativity, appeared to have dealt the concept of time a death blow from which recovery was impossible. He famously wrote, following the death of his friend Michele Besso and shortly before his own death in 1955, “Now he has again preceded me a little in parting from this strange world. This has no importance. For people such as us who believe in physics, the separation between past, present and future has only the importance of an admittedly tenacious illusion [1]”. In stating this, was Einstein simply trying to assuage the grief of his friend’s family with supportive words, or was he stating his literal belief? Many physicists today would take his expression literally [2]. The universe is eternal; all things past, present, and future co-exist, and therefore, so do we. Change does not occur; it too is an illusion, merely a statement that there are differences between different elements of the universe [3]. Nothing actually happens, and thus, fundamentally, there are no causes because no event can bring about another event since both co-exist. Ideas of precedence and consequence are meaningless. Relations between different elements of the universe are merely correlational, possibly expressing some underlying symmetry or possibly merely spurious. Indeed, within physics, force, which is associated with the concepts of agency and action, gives way to energy, which is an expression of symmetry and is primarily relational.

For those who ascribe to a static, timeless view of reality, there is a fundamental problem: “how to account for the appearance of passage or temporal becoming without presupposing the becoming of the appearance [4] (p. 14)”. If time is an illusion, then it is an illusion that is experienced *in time*. There is simply no other way to experience it.

Physics has focused on the study of inanimate matter, but inanimate matter is passive, lacking any aspect of agency or intention. The motion of inanimate matter is accidental and not an intrinsic or intentional property. Inanimate matter cannot determine its own motion. Motion is determined by interaction, which may be direct (e.g., collisions) or indirect via fields (e.g., gravitation), and is most often described in terms of kinetic and potential energy. Inanimate matter cannot intentionally regulate those fields or, therefore, their motion.

Both the existence and motion of inanimate entities are most commonly depicted as continuous curves (trajectories) in a space–time diagram, akin to lines in a painting. Trajectories must be indexed relative to the entity being represented, and continuity expresses the enduring nature of the entity. Enduring existence is denoted by a non-trivial trajectory, while motion is denoted by trajectories that are not parallel to the “time” axis. Segments of trajectories are assigned to symmetry classes (energy, momentum), and interactions result in a change in symmetry classes among segments [5]. Remarkably, these changes appear to be “lawful”. These segments are, in general, regions free of interactions. Nothing actually happens in such a picture, just as nothing actually happens in a painting. As a representation of events, there is no problem. However, according to those who espouse a block universe conception of reality, this space–time diagram is not just a representation of reality; it is an *exact* representation of reality (somewhat like the difference between a painting and a photograph). Everything that was, is, and will be co-exists in this block universe, just as it does in the space–time diagram. There is no time. Nothing *happens*; it simply *is*.

The biological world, however, stubbornly refuses to accept the non-existence of time. Organisms are born, live lives, reproduce, and die. Unlike inanimate matter, organisms are agents: they act upon the world, modifying their surroundings to meet basic needs for shelter, for food, for reproduction, and for defence. Their behaviour is intentional, purposeful, variable but not random, and expresses some functionality. Organisms extract salient information from their environment and generate actions that have ecological and ethological significance and relevance.

If time is merely a psychological illusion, then why do organisms (including single cell organisms such as amoebae, which have no pretence of psychology) act as if it were real? If everything co-exists, then why should there be birth, ageing, and death? If there are no causes, then why are there symmetries at all in the universe? Why is it not merely random?

Einstein’s choice of words in his statement regarding the illusory nature of time are more revealing than might appear at first glance. Let me repeat it. “For people like us who believe in physics, the separation between past, present and future has only the importance of an admittedly tenacious illusion”. He appears to be saying that his belief in physics provides sufficient authority for him to have knowledge of something. It is not clear, at least at this stage in our methodologies, whether it is possible to determine whether the universe does or does not behave in accordance with the presence or absence of time. Therefore, one can have no *direct* knowledge of the existence or absence of time. The empirical side of physics cannot have such direct evidence since, if it did, so would we. So, it must be from the theoretical side of physics that such knowledge is obtained. Yet, how can *theories* about something serve as *knowledge* about that something? A theory is a conceptual entity, conceived in the mind of someone, imparted to and elaborated upon by others, codified in some manner, whether in common language or the language of mathematics, and then utilised to provide understanding, to guide our actions, to make predictions, and so on. A theory is a product of the mind, not of the entity to which it pertains. In the absence of direct evidence, theory may attempt to substitute itself in place of that missing evidence. However, this substitution of theory requires belief, and belief must be grounded in some manner that determines the confidence with which we hold the belief. Faith requires belief, but the grounding is in an appeal to authority. Belief in science is said to be different, as it is grounded in empirical evidence. However, if there is no such empirical evidence, then what grounds the belief? In the case of theory in physics, the grounding of belief would appear to rest in the logical consistency and coherence of the mathematics that is used to express the theory. However, there is an infinite amount of mathematics potentially available; what determines the choice of the mathematics to be used?

In this paper, I argue that the denial of the existence of time is determined by a particular worldview, a worldview that is so pervasive that it has become implicit, resulting in an implicit bias in the manner in which we conduct physics. In my view, science is *not* about the promotion of a particular worldview. Rather, science is a collection of methods for interrogating the natural world in an attempt to discover knowledge about that world, its entities, their interactions, and their causal relationships, in which we can hold the greatest confidence. I argue that once this implicit worldview is made explicit, then the majority of arguments purporting to prove the non-existence of time are shown to be examples of circular reasoning, of “begging the question”. Far from proving the non-existence of time, I argue that physics is actually agnostic on this question, and it will remain so until the stage comes (if it ever comes) at which it is possible to obtain direct empirical evidence for or against the existence of time.

## 2. The Importance of Worldviews

Implicit bias is a well-recognised phenomenon that, if not acknowledged, can cause individuals to make implicit assumptions and errors of reasoning, which can lead to erroneous or even harmful conclusions. The existence and influence of implicit bias are widely recognised within psychology, psychiatry, sociology, and anthropology, where steps are taken to try to detect and mitigate its influence [6]. Implicit bias has not been discussed as much in physics or mathematics. Unger [7] argued that an important role of philosophy is to make apparent the hidden assumptions and biases that shape the development of a theory or a discipline. Failure to take notice of these hidden assumptions can be a source of paradox, confusion, and inscrutability. The scientific method has served well over the centuries to reduce the impact that implicit biases may have on the formation of hypotheses and theories. Empirical disputation or validation provides a powerful tool to challenge ideas, and the willingness to accept empirical results over theoretical arguments has helped to reduce the effects of implicit biases. However, when empirical evidence lacks, the possibility of implicit bias influencing theory formation becomes significant. Scientists do not operate in a vacuum, and there are sources of bias that are so deeply ingrained that they are simply accepted as unquestioned truth. Indeed, in some fields, such as high-energy physics, the value of a theory often has more to do with its mathematical value than its connection to empirical evidence [8].

Inanimate matter has no need for worldviews. Being incapable of acting, it has no need to know what it can or should act upon, in what manner it should act, for what purpose it should act, or how to judge the effectiveness and consequences of its actions. Organisms are entirely different. For them, answers to these questions may be essential to their survival. These answers are provided in the form of a worldview. A worldview may be explicit (such as expressed in culture) or implicit (encoded in its dynamics).

The phenomenon of transient-induced global response synchronisation (TIGoRS) suggests one mechanism by which one aspect of a worldview, that of determining salience, might be encoded implicitly in the dynamics of a complex system [9,10,11,12]. TIGoRS has been observed in a number of complex system models, including cellular automata, coupled map lattices, tempered neural networks, dispositional cellular automata, and cocktail party automata [9,10,11,12]. A cocktail party automaton is an inhomogeneous finite cell (*N*), a finite state set (*S*), and an adaptive cellular automaton. The local updating rule changes for each cell according to its own local neighbourhood. The automaton is configured in some initial state in which the local rules and state values for each cell are assigned and then allowed to evolve autonomously, resulting in a variety of complex spatio-temporal patterns of states and rules.

A fixed spatio-temporal pattern (dimension N×T for some duration *T*) is created, with the value at any point being a possible state value from the set *S*. The automaton is allowed to undergo its natural evolution, and simultaneously, a random sampling of this pattern is carried out. Suppose that the automaton has evolved up to time t−1. Then, a random sampling of the pattern at sites (,t) is undertaken. If the pattern is sampled at site (n,t) with value *v*, then the state value of the automaton cell *n* at time *t* is given the value *v*. The states of all other cells are given according to their natural evolution from time t−1. The process is repeated up to time *T*. The N×T spatio-temporal pattern generated by the automaton is then compared to the original pattern. A sampling rate of as little as 10% can result in a 95% match (measured by Hamming distance) between the output of the automaton and the original pattern. This response is stable over a range of initial configurations, yet, interestingly, the final rule configuration may be different for each initial condition. The set of patterns that are capable of eliciting TIGoRS behaviour is restricted. Patterns that do not elicit TIGoRS result in outputs that appear random.

Thus, from the space of all possible input patterns, the automaton responds to a select few by reproducing the pattern, which could then be passed on to other automata for further processing. Patterns that are not from this collection result in a noisy output, which would serve merely as a perturbation to downstream automata. In effect, the automaton parses the space of possible patterns into those that are salient (via TIGoRS) and serves as a gate, permitting only information relevant to those salient patterns to be propagated to downstream automata. Thus, the cellular automaton *embodies* this aspect of its worldview through its dynamics, implicitly and without the need for any explicit representation. A consistent theory of dynamical systems with dynamical transients as the central entity for study includes dynamical automata [10,11] and Sulis machines [13].

Worldviews are embedded in cultures, which ascribe meaning to individual behaviours, set norms for acceptable and unacceptable behaviours, suggest which behaviours are appropriate for which situations and so on. The failure to appreciate the conditionality of worldviews has led to much conflict over the millennia, as has the damage that has occurred when one worldview is dogmatised and weaponised. Worldviews also appear in the arts, in philosophy, and, most importantly for this paper, in science.

The concept of a semantic frame [14] was introduced many years ago as an attempt to provide a formal representation of this idea of a worldview. The semantic frame is a conceptual entity that encodes, in some manner, answers to the six main epistemological questions:*Who* are the relevant entities with which to interact?*What* are the relevant properties/characteristics of these entities?*How* are interactions structured, determined, and expressed?*When* do these interactions take place?*Where* do these interactions take place?*Why* do these interactions occur: their cause, their functionality, their ethological relevance, their purpose, and their relevant intention?

Worldviews underlie the conduct of science just as much as they do in everyday life. Worldviews reflect (and eventually determine) which natural entities should be the focus of scientific study. They reflect (or determine) the manner in which these entities are to be observed or interacted with. They reflect (or determine) the kinds of questions to be asked, and how to interpret the results that are received. Within an overarching worldview, there may be more restricted worldviews pertaining to more circumscribed domains of inquiry. For example, in particle physics, a common practice is to collide particles with one another at very high speeds. This is not a typical practice in cell biology or anthropology. Pickering [15] referred to this as the “mangle of practice”, expressing the understanding that science is not merely a collection of ideas or theories, but a collection of embodied practices by means of which we directly interact with the natural world.

These practices, when they become standardised, begin to form a worldview. This can have profound consequences. For example, Lejarriga and Hertwig [16] have shown that the use of different experimental paradigms (the equivalent of semantic frames) has produced strikingly divergent results in decision theory, leading to two opposing views of human reasoning: one that we are fundamentally rational but sometimes commit errors, and one that we are fundamentally non-rational but sometimes manage rationality. Biomedical science has run afoul of these kinds of worldview-driven biases [17]. For example, the long-standing bias against the use of female subjects in experiments (due to presumed greater unpredictability or complexity in responses) has been contradicted by a recent study [18] showing that, in fact, it is males who exhibit greater unpredictability in behaviour depending upon context. This may have profound implications for the interpretation of the results of such experiments and their use in clinical medicine.

The remarkable effectiveness of mathematics in physics and the close relationship between their developments have been noted for a very long time. Prior to the adoption of a logic-based, axiomatic approach in the 19th century, mathematics had much more of a philosophical, exploratory air about it than the rigorous structure that exists today. Older mathematics used to be readable, compared to the dry succession of theorems and proofs seen today. The entities of mathematical thought have grown in number but are never extinct; once discovered, they seem to be eternal. The body of mathematical knowledge is held together by ideals of truth (and falseness), with its propositions being either eternally true or eternally false. Mathematicians are, however, guided by more than logic. Aesthetics play a role as well. Wigner [19] begins his famous paper on the unreasonable effectiveness of mathematics in physics with a quotation from Russell:

“Mathematics, rightly viewed, possesses not only truth, but supreme beauty cold and austere, like that of sculpture, without appeal to any part of our weaker nature, without the gorgeous trappings of painting or music, yet sublimely pure, and capable of a stern perfection such as only the greatest art can show. The true spirit of delight, the exaltation, the sense of being more than Man, which is the touchstone of the highest excellence, is to be found in mathematics as surely as in poetry.”

There is something of the divine to be felt in Russell’s depiction. Mathematics appears as an embodiment of Plato’s ideals, untarnished by the vicissitudes of a natural existence with all of its messiness. The entities of pure mathematics, together with its practices, exemplify what I will call the objectivist worldview (not to be confused with the Objectivism of Ayn Rand [20]), formally expressed in the objectivist semantic frame. The entities of the objectivist worldview are *objects*. Ideal objects, collectively, possess several essential characteristics: they are eternal, statements about them are timeless, their properties are intrinsic (and also timeless unless disturbed in some manner), and they embody or express some kind of ideal often expressed as being symmetrical, optimal, rational, extremal, canonical, and universal. Statements about objects are logical propositions; thus, they are either eternally true or eternally false. Structures and relationships between objects are governed by logic, and knowledge concerning them can be acquired through logic (although intuition, guesswork, and play are also important in actual practice) [21,22]. These points have been emphasised by Unger [7]. The objects of mathematics are isolated from the natural world and can be isolated from one another. They are ideals, and pure mathematics provides the purest expression of the objectivist worldview.

The worldview of physics has been strongly influenced by its long association with mathematics. The principal entities for study in physics are forms of inanimate matter, which happen to come closest to the ideal of a mathematical object. Inanimate matter can be isolated, and thus rendered effectively eternal; its properties can be described in propositional terms, present or not present, true or false; its behaviour appears lawful, rational, and logical. Its structures have inspired generations of mathematicians, especially in geometry.

The use of mathematics in physics has generally been referred to as that of a language, but this ignores the deeper connection between them, namely, the sharing of a worldview: the objectivist worldview. Mathematics possesses a dual character: it is both a language for describing relations between entities, and the subject to which that language is applied. These two aspects are inextricably intertwined and inseparable. Individual mathematicians (and physicists) are free, of course, to be guided by any worldview of their choice when *applying* mathematics to a problem. Nevertheless, the mathematical entities themselves are objects within the objectivist worldview, and there is no way to use the language of mathematics without being influenced (implicitly or explicitly) by the worldview associated with mathematics as a subject. This is especially true when the results of mathematical arguments are to be interpreted in relation to natural kinds, and not merely serve as calculations. It is hoped that the days of “shut up and calculate” are long behind us, especially since foundations research recently won a Nobel Prize. In order for a worldview to be effective (and, by interpretation, meaningful), it is essential that the entities to which it is applied are homologous (to a significant degree) in their characteristics and behaviours with the characteristics of entities as presented within the worldview.

A major impetus for introducing the concept of a semantic frame was to try to understand what distinguished simple systems from complicated and complicated ones from complex. Simple and complicated entities are distinguished primarily by the number of interacting entities. Complex, on the other hand, refers to systems or situations that admit descriptions by multiple mutually irreducible semantic frames, usually applied to subcollections or subsystems rather than to the whole.

An accessible formal example of this is Conway’s cellular automaton, “Life” [23]. Life is a two-dimensional, two-state cellular automaton. The number of cells is arbitrary but finite and usually arranged in a square lattice of $N^{2}$ cells, where *N* is the number of cells along one side of the lattice. The two states are 0.1. A neighbourhood of a cell consists of the eight immediately adjacent cells. Let *X* be the sum of the states of the cells in this neighbourhood. The states of the cells are updated simultaneously. The rule is that if the state of the cell under consideration is 1, then it remains 1 if X=2,3; otherwise, it becomes 0. If it is 0, then it remains 0 unless X=3, in which case it becomes 1. One can choose any initial configuration of states and use the updating rule to evolve the automaton forward in time. The description above provides us with the basic semantic frame, the cellular automaton frame.

Conway showed, however, that there exist particular local configurations of states and particular arrangements of these configurations forming larger organisational patterns, such that, with suitable interpretation (Turing semantic frame) the subsequent evolution of these patterns expresses in an exact manner the behaviour of a universal Turing machine, meaning that it can emulate the actions of any computer. One can provide an initial configuration of the automaton, arranged in patterns consistent with this interpretation, which can be interpreted as the initial data of a computation as well as the configuration of a Turing machine (computer), whose subsequent evolution will correspond exactly to the computation and whose final configuration can be interpreted as the outcome of the computation. This is only possible if the initial configuration and its subsequent evolution are consistent with this interpretation. The role of the semantic frame is to ensure that our interaction with the automaton remains consistent with its interpretation. Note that the Turing frame applies only to a subset of possible configurations of the automaton. There are a potentially infinite number of configurations that are inconsistent with this frame. They may well be consistent with other possible semantic frames, but research regarding that is currently lacking. The Turing frame does not describe all possible evolutions of patterns in the automaton. Likewise, the basic automaton frame, while necessary to evolve the states of the cells in general, does not provide the information in and of itself to specify the Turing frame.

It is naive, or perhaps disingenuous, to assert that a single frame or worldview, such as the objectivist frame, serves to capture all that is complex within the universe. Even if the objectivist worldview of physics is foundational, such as the cellular automaton semantic frame, it may be just as incapable of describing the whole panoply of emergent systems that arise, which will require irreducible semantic frames such as the Turing semantic frame for their description. There are indeed other semantic frames that apply to a wide range of phenomena. Three of them, objectivist, subjectivist, and processist, are presented in Table A1. Four broad classes of entities and their characteristics are presented in Table A2. I will return to the processist worldview in the third section.

An important characteristic of the objectivist worldview is that its statements are like logical propositions: timeless, a point emphasised by Unger [7]. There is no place for ideas such as becoming for happening or for history. It is not surprising that viewing the world from a worldview in which time is irrelevant might lead to arguments and conclusions that time does not exist. This will be critically important in understanding how metaphysics, the philosophy of science, and physics all come to the conclusion that time does not exist. In the next section, I show how arguments used in philosophy and physics to prove the non-existence of time contain within them the implicit assumption of such non-existence, begging the question and thus rendering the argument circular. I will argue that the idea of “no time” is not a logical consequence of our theories but rather an implicit assumption of them, reflecting an implicit bias towards object-based metaphysics.

Before proceeding to examine the arguments for and against the reality of time, it is first necessary to address the question of what time is being referred to. Time is not a unitary construct, and there are multiple competing conceptions of time. When people speak for or against time, most often they are speaking for or against one specific conceptualisation of time, not the concept of time in totality. The major features of time to be considered are:Time as a physical entity or time as a relation.Time as absolute or time as emergent.Time as a background or container, or time as a relation.Time as moment or as duration.Metrical time versus ordinal time.Time as static or time as flowing.Time as change or change in time.Being or becoming.Universal time or parochial time.Local time or global time.Reversibility of time or reversibility in time.Reversibility versus irreversibility: locality versus universality.

In this paper, I cannot do justice to all of these different aspects, so I shall focus on just one question: Does our current understanding of physics truly prove that time, in the sense of becoming or of happening, cannot exist? In other words, does physics preclude the possibility of *becoming* and only allow for *being*? Arthur has presented a detailed and deep argument in favour of the existence, indeed the necessity, of local becoming [4]. I shall not repeat his argument here. Rather, I will argue that the implicit influence of the objectivist worldview on the framing of arguments for the non-existence of time amounts to begging the question. The arguments are rendered circular and without substance. In the interest of length, I shall not address every argument that has been put forward against the existence of time, but focus on those that are most frequently cited in the literature.

## 3. Arguments against the Existence of Time

In this section, I shall examine arguments from philosophy, relativity theory, and quantum mechanics purporting to prove the non-existence of time. There are three key fallacies that I shall refer to repeatedly throughout this discussion. The first is Whitehead’s fallacy of misplaced concreteness [24]. Mermin’s notion of reification [25] is an example of this. The fallacy of misplaced concreteness is invoked when, in an argument, one conflates or identifies an abstract construct or entity with a concrete entity. A simple example is identifying an entity with its description and attributing the properties of the description to the entity being described. The second is begging the question. This occurs when the framing of an argument explicitly or implicitly assumes the truth of the conclusion. Such an argument amounts to proving A is A; true, but not useful. The third is what I have called the Fallacy of Misplaced Omniscience. This fallacy is invoked when an appeal is made to knowledge that cannot be obtained except by virtue of the vantage point of an observer that exists outside of the system being discussed; a vantage point that cannot be reached by any entity within the system. This kind of argument is commonplace in mathematics, but it is not fallacious because mathematics is conceptual and ideational, existing in the mind of the mathematician who conceives it. The mathematician is free to assume an omniscient vantage point because they are effectively the creator and they stand outside the system being studied. However, when applied to situations in which the observer of a system is also a component of the system, one cannot assume the existence of knowledge that does not actually exist in order to prove an argument. This amounts to attempting to prove that A⇒B when, in fact, one has ¬A in hand. That amounts to stating A∧¬A, which can then be used to prove anything.

The existence of time is inextricably linked to the existence of change and causation. Consider a universe with just a single unchanging entity. In such a universe, there is neither space nor time. There is no space and no inertial motion because there is no reference point by which to judge two positions of a single entity to be different. There is no time because if no change takes place between two presumed times, then, by any measure, there is no discernible difference between the entities at these two times, and so, by Leibniz’s principle of the identity of indiscernibles, they must be identical. Therefore, there can be no difference in time.

Space requires at least two distinct entities for it to be discernible, while time requires an entity to manifest at least two distinct states for it to be discernible. This suggests that space and time are *relations* between entities; they do not exist in and of themselves since there is no means to know of them unless there are at least two distinct entities to mark a difference (in space) or at least two distinct states to mark a difference (in time). Thus, space and time are not material entities in their own right but rather represent relations between entities.

If space and time were material entities, then how would motion and change take place? In [26], I suggested that a basic requirement for something to be *real* is that it makes some difference. For the materiality of space and time to be real, by this criterion their materiality would have to make a difference. Wheeler famously wrote, “Spacetime tells matter how to move; matter tells spacetime how to curve” [27], which suggests that space and time do make a difference. However, in general relativity, space–time exists as a complete whole: nothing actually happens and nothing does anything. The equations of general relativity describe a relationship between certain geometrical features of a space–time representation and certain intrinsic (physical) features of the entities that the diagram does not directly represent (such as mass, charge, and so on). These geometrical features are relational, not material. The relationship may be real, but a space–time point itself cannot be measured without an entity being present at that point to mark its existence. The entity is real, and the relationship is real, but the space–time point has no independent reality.

Arthur makes a similar assertion: “Time is not a physical *thing* like a river. Nor is time itself a *process* that can flow at a certain rate, since if this is taken literally we would need a further time against which to measure this rate... Time does not actually *do* anything, but anything that *does* do something, does so in time. This is what we call the passage of time... *It is time flow in this sense of passage, the reality of temporal becoming*” (SIC) that Arthur argues for [4] (p. 1). Arthur goes on to present his case for the reality of local becoming.

When physicists deny the existence of time, they do not deny the existence of change; rather, they deny the existence of becoming. Returning to the painting metaphor, different locations on the canvas are marked with different paints. Change is simply the assertion that this is the case. Yet, nothing happens; these colours are there for all time. The only way to observe such change is by standing outside of the painting, invoking the Fallacy of Misplaced Omniscience. Within the painting, there is only stasis; these differences are unobservable. Yet, then, how is consciousness supposed to register these differences, thereby creating the illusion of time passing, if consciousness is an attribute of us and we are in the painting in which all is static?

The central issue is not whether change may be discerned, as in being a discernible difference between two entities or two events (such as the painting), but rather whether change is something that *happens*. If something moves, then at one time *it was* in location *A* and at another time *it is* in location *B*, and *it* traversed at least some of the locations between *A* and *B*. When *it* is at location *A*, *it* is not at location *B*, and *it* is at location *B* only when *it* is at location *B* and not at location *A*. Afterwards, it is at neither location *A* nor *B*. Physicists argue that change can exist in the absence of happening, but does physics truly prove this to be true?

### 3.1. Philosophical Arguments: Zeno, McTaggart, and Grounding

Let us begin the discussion by considering the most cited philosophical arguments against the existence of time. First, though, recall for reference the essential properties of an object, the entity of the objectivist worldview. The core attributes of an object are:It exists independently of any other entity: it can be isolated and treated as a whole unto itself.It is eternal: it does not *become*, it merely *is*.It is passive: it reacts, it does not act.Its properties are intrinsic and non-contextual: they are fixed, complete, and independent of the actions of any other entity.Its motion is determined by fixed laws, which may be deterministic or stochastic (usually explained away due to ignorance on the part of the observer).Its motion is often attributed to variational principles—optimality, minimal, and maximal—always extreme in some direction.Its interactions with other objects are always local.History is irrelevant: the future motion of an object depends only on its present state (and sometimes not even that, in the case of stochastic objects).

Among the earliest philosophical arguments are those of Zeno. In his dichotomy paradox, an entity is to travel from location *A* to location *B*, a distance of *d*. First, it must move from *A* to a point at a distance d/2 from *B*. From that point, it must move a distance d/4 towards *B*. From that point yet again, it must move a distance d/8, and so on. No matter how close it may get to *B*, it never reaches *B* because it must always traverse one half of the remaining distance first. Greek philosophers had already conceived of the concept of point and, with it, the notion of denseness. Time became conflated with the rational numbers used to measure it, and the idea of “now” became identified with that of an ideal point: an example of the fallacy of misplaced concreteness.

In his arrow paradox, Zeno argued that if a moving arrow occupies a definite position at a time t, then this situation is identical to an arrow being at rest in the same position at time t, and therefore the moving arrow must be considered to also be at rest at time t. Yet, then, it must be at rest at every subsequent time t’, and thus it is always at rest and no motion occurs. Although Aristotle denied the possibility of motion at an instantaneous now, he did allow for the possibility of motion across an interval. According to Arthur [4] (p. 23), Aristotle argued that Zeno committed a logical error, the fallacy of composition, in which he assumed that time is composed of nows (again in the sense of ideal points).

Zeno’s argument, however, assumes the identity of “now” with “point”, the latter being an objectivist concept. Yet, then, the argument that “nothing can happen at a point; therefore, nothing can happen now, so now cannot exist” simply begs the question.

Arthur writes of Aristotle’s conception of now [4] (p. 23) “An indivisible for him is an endpoint of a magnitude or interval. Thus a point is the indivisible beginning or endpoint of a line, and a *now* is the indivisible beginning or endpoint of a stretch of time. There can be motion across a stretch, but not at an instant or now. Thus, for Aristotle “nothing moves in the now”” (*Physics* 234a 24); but, by the same token, “nothing can be at rest in the now” either (*Physics* 234b 7)(SIC). This suggests that now might be better understood as an interval. Indeed, the real number system is constructed by equating real numbers with “cuts” or boundaries between certain pairs of sets of rational numbers. Unlike rational numbers, which are constructible, real numbers have an air of unreality about them since they can never be explicitly denoted. They only exist in the sense of endpoints: the nothing between somethings. Yet, it is to this boundary that is assigned the quality of number. Now becomes associated with these nothings, while the somethings, the durations, are ignored. The association of time with these nothings pretty much renders arguments for its non-existence a foregone conclusion. However, it is the interval, the duration, which is the now.

Psychologically, the experience of now is not instantaneous; it has a duration [28]. Organisms do not respond to instantaneous events; stimuli always have some (small) duration. Even physical systems do not actually respond to instantaneous events. That is an artefact of the use of continuous mathematics to describe physical systems. In the photoelectric effect, the ejection of an electron from a metal upon illumination by a beam of light depends on the energy of the incident light, not its intensity. While intensity can be detected in an instantaneous manner, energy cannot. Energy is determined by the frequency of light, and frequency cannot be determined by sampling at a single point in time. It requires at least two points separated in time by at least one half of a cycle.

Every event has a beginning and an ending. Between the ending of one event and the beginning of another, there is an infinitesimal moment. It is this moment that can be mapped to an element of a real one-dimensional manifold. It is this moment that can be said to mark an instant of time. However, note that in such a moment, *nothing happens*. Such a moment is devoid of content. It is nothing, but it is a nothing that marks the boundary between two somethings. At this moment, the preceding event can be considered to be complete, whole, manifest, and realised, and this moment may be utilised to mark the occurrence of the event. However, the event itself did not *happen* in the moment, it happened throughout the duration that preceded the moment. Prior to that moment, the event is happening, and, although incomplete, the observing mind may anticipate and predict its completion and begin planning its subsequent actions based upon that prediction [29]. Subsequent to this moment, the next event begins to take form. Thus, even though an event is a temporally extended entity, when we attempt to assign a *time* to it, the only time that can be consistently assigned is the instantaneous time that marks its completion. Such a conception of time is perfectly rigorous mathematically. After all, it is simply the idea of the Dedekind cut, which is used to define the real numbers in the first place, applied to the collection of events. Positioning an event in time via its termination does not lead to integer or rational time as a necessity, any more than do the Dedekind cuts. The relationship between one such cut time and another may be integer, rational, or real, depending upon the event that separates them.

TIGoRS demonstrates a phenomenon for which duration is a necessity. The concept of dynamical automaton shows that a dynamical system predicated upon transients is just as viable a model for dynamics as one based upon instantaneous states and events [12]. The attachment to the concept of infinitesimal time and to the now as a mathematical point reflects a philosophical prejudice, an implicit bias for the objectivist worldview, which pervades Western thinking on these questions. The idea of time as duration leads to a form of presentism that is unlike those commonly discussed in the philosophical literature. There is no present *moment*. Nothing happens at a moment, neither motion nor change. Motion and change happen only during durations. There is no reason, a priori, that durations should align themselves in any formal pattern; thus, there need be no universal *now*, no universal simultaneity. The universe can be a collection of durations, corresponding to the different events that happen in it. As Arthur states, each event happens, or exists, precisely at the time that it happens and not at any other time. It comes into being and fades away asynchronously when compared to other events. The events corresponding to the history of an entity can be ordered in relation to their becoming, but the relationship between that ordering and any other is not pre-determined and depends upon the flow of information between entities. In [5], I argued that information is the main determinant of behaviour in complex adaptive systems. It is also a major determinant of the behaviour of quantum systems.

One of the most widely cited modern philosophical arguments for the non-existence of time is that of McTaggart and his famous A, B, and C series. McTaggart’s A series orders events relative to a particular observer and a particular now, ordering events as past, present, or future relative to that now. This now thus possesses indexicality or token-reflexiveness; it must be referenced to the time at which the statement concerning positions along the A series is uttered. Since now changes, these attributes of events must change, making them accidental and thus not intrinsic to the events themselves. These changes in attributes must themselves be taking place in time, but this cannot be the same time that the now describes, and so an auxiliary time is needed, leading to an infinite regress. This renders the A series meaningless. However, note that in this formulation, McTaggart implicitly assumes that events are objects, and therefore statements about attributes must have a propositional character. This assumption represents an implicit bias for the objectivist worldview: events, and statements concerning them, are implicitly assumed to have a timeless quality, so a tempered representation cannot hold and therefore must not exist. Again, this begs the question.

The attribution of tense to an event is not a property of the event but rather is a projection of the observer onto the event based upon their own history up to the time at which the observation is made. It is a subjective statement of understanding that has truth in the moment of observation and not at any other moment. It is not a proposition (i.e., it does not possess a timeless truth or false valuation). Thus, it is not an objective statement. To assume that only statements and entities that conform to the objectivist worldview are “real” is an act of dogmatisation, not an act of scientific practice. A similar bias appears in decision theory, where only rational decisions are accorded value (i.e., decisions that are supported within an objectivist worldview). To say a decision is non-rational does not make it false, spurious, illogical, fanciful, delusional, or irrational; it simply makes it not rational. Organisms never achieve the conditions needed to carry out ideal rational decisions; they come close sometimes, but at other times they settle for decisions that are merely “good enough” [30,31]. To declare otherwise exposes an implicit bias in favour of the objectivist worldview, and to persist in enforcing it is another example of dogmatisation.

The B series is a precedence relation, which says that one event is either earlier than, simultaneous with, or later than another event. The idea of the now appears to be eliminated from this series. It is a timeless relation to all events and thus clearly an objectivist concept. However, the B series does make reference to time. If events come into existence, then how does one determine the order of all of the events that are held to be later than some event? If they do not already exist, then they must lie about the future of what does exist, and hence one must invoke the A series. However, it has supposedly been found meaningless. Hence, in order to form the B series, all events must co-exist, but this means that, in order to assume the existence of a B series, we must assume that the universe is a block universe, and so time as becoming does not exist. Again, we beg the question.

McTaggart suggested the existence of a third series, a C series, in which there is no reference to time yet events still possess some kind of intrinsic ordering. Yet, again, if one creates a series without time, it cannot be used to show that time does not exist; more begging the question.

As Arthur [4] points out, the assumption of a block universe presents its own contradiction, because if all events exist and neither become nor fade away (which would imply change and thus time), then they must persist, but the notion of persistence itself implies persistence in time. Thus, there must be some time to which the persistence of these eternal events must be referenced. Yet, the existence of time is denied.

There have been many arguments for or against time that focus on the existence or non-existence of tensed propositions or statements, whether in logic or linguistics [32,33,34,35,36,37,38]. I will not review them here since they appear to be attempts to apply objectivist worldview constructs, particularly propositions, which are by definition timeless, to tempered phenomena. This seems akin to asking a blind man to describe the changing colours of a cuttlefish. I will examine one prominent argument against presentism: the grounding problem. Mozersky views the grounding argument as a reductio ad absurdum argument, which I quote [39]:“There exist determinately true and false propositions about the past case.Truth supervenes on what exists.What exists in the present underdetermines what is true in the past.All and only that which is present exists.Therefore, there are no determinately true or false propositions about the past.”

Mozersky presents several defences of presentism against the grounding problem (referring to the non-existent, tensed properties, ersatz B-series, Haecceities, evidence, and quasi-truth), and argues that each of them fails to refute one or more of the statements above. In general, these arguments hinge upon the propositional nature of these statements, which, by definition, makes them timeless. Implicit in these statements is timelessness, and so using them to prove time does not exist is another example of begging the question.

Consider the pair of statements:There exist determinately true and false propositions about the past.Truth supervenes on what exists.

The first statement runs afoul of the Fallacy of Misplaced Omniscience. There is no reason whatsoever that a determinately true or false proposition about the past should be able to be determined by us. There is no reason that everything that can, in principle, be known about the past should be known to us. To assert otherwise places us in the position of an all-knowing, god-like being. Making the same assertion about the future is even worse because it means that we must be able to apprehend the entire history of the universe. The second statement is an objectivist statement. Note the use of the present tense. There is no need for an entity to exist when a statement about it is made. It is only essential that the statement refer to an entity that did exist prior to its utterance. Whether the truth value of the statement can actually be determined depends on the existence of supporting evidence. Mozersky rejects the evidence-based arguments, but when I utter the statement that Napoleon was defeated at the Battle of Waterloo, I do not need to travel to Europe, to the location of the battle, or to the time of the battle to directly witness Napoleon being defeated. To assert otherwise is absurd and would repudiate the practice of historical research. Statements about the past or the future cannot be propositions: their truth value is always conditional. To assert otherwise implies an objectivist worldview and invokes the Fallacy of Misplaced Omniscience, for how else can one know the entirety of reality.

The statement “What exists in the present underdetermines what is true in the past” is a true statement, but it does not underdetermine what is true of the past in an absolute sense (after all, there might be evidence existing elsewhere that will someday be discovered), but only in a relative sense that at the time of uttering a statement, the actual truth value cannot be determined by us. Moreover, it does not mean that the underdetermination of the past by the present implies that the past itself, in its actuality, is somehow underdetermined. It simply means that meaningful information is lost over time. A more accurate expression of the statement would be “What exists in the present underdetermines what is known about the past”.

### 3.2. Classical Arguments: Temporal Reversibility

Another line of argument for the non-existence of time is based on the time symmetry of the equations of motion. Time or symmetry (or time reversibility) means that the equations of motion remain invariant under the time reversal transformation t→−t. Hence, if f(t,x) is a forward solution of the equation, then f(−t,x) is a solution of the time-reversed equation. This simple transformation, however, has evoked numerous competing interpretations. The *ultimate* version is that it is possible for time *itself* to be reversed, so history could unfold like a movie played backwards. The *strong* version allows a system to travel to a spatio-temporal location in its past light cone. Once in the past, the system advances forward in time like everything else, except it is now present in the past when previously it had not been. If the universe is a block universe, then how does one *travel* back in time if there is no time and no motion.

The *weak* version (advocated by Hardy [40]) merely asserts that if it is possible for a particle to travel from location *A* to location *B*, then it should also be possible for it to travel from location *B* to location *A*, if all other conditions remain unchanged. One can hardly quibble with that interpretation.

The assertion of both the ultimate and strong versions requires that the universe be a block universe because the ability of any entity from the future to be able to travel to any point in the past requires the co-existence of all past and future events. However, that is an expression of the objectivist worldview, and any argument using this form of temporal reversibility to argue for the non-existence of time in the form of becoming must beg the question. Moreover, as Arthur points out [4], to travel along a space–time path would mean moving with respect to some time, but since time is already included in space–time, there is no other time to which such motion could be referred. Even if there were, it would fall prey to the argument against time used to justify the block universe model in the first place.

In a universe governed by becoming, by transience, every spatio-temporal location within the past light of an entity ceases to exist and certainly would not exist at the time that the system attempts to travel to the past. In a universe with becoming, time travel is impossible.

Note that temporal reversibility, interpreted in the weak sense, is compatible with either a block universe (an objectivist frame) or becoming. Arthur [4] and Unger and Smolin [7] have all argued that becoming and (weak) temporal reversibility are compatible.

In an objectivist block universe, the notion of a temporal or causal order becomes meaningless. Since everything *is*, to what does this order refer? If there is temporal reversibility, then it is impossible to know which temporal ordering is the *true* ordering. Any order becomes as good as any other.

Let us return to the ultimate version. I have argued that invoking it to deny the existence of becoming is an example of begging the question. The equations of motion in both classical and quantum mechanics are *local*: they specify the relationship between various derivatives and the values of certain functions at single points. To assert that the equation holds universally runs afoul of the Fallacies of Misplaced Omniscience and concreteness. There is no means to confirm it unless one can examine every location, past and future. Appealing to a block universe fails because measurements of this kind must take place in time, but since time is already incorporated into the block universe (the associated space–time diagram), then what is this additional time [4]? Although the equations of motion are local, they are meant to be applicable at every space–time location in the base space. To solve these equations, it is necessary to fix initial and boundary conditions, which are generally *global*. Missing from the equations is any expression of *history* that is to be found, in part, in the initial and boundary conditions, which must be chosen by whoever applies the equations. In mathematics, these may be chosen by appeal to the mathematician’s divine position, but to a user embedded within the system being described, these conditions are either informed guesses or must invoke some degree of block universe.

But there is another problem. Consider the simple case of a billiard ball sitting at rest on a table. If one tries to evolve the equation of motion for the ball backwards in time, it will remain at rest. Its initial motion is thermal, and time reversed thermal motion is still thermal. Yet, the ball has not been there for all eternity. To evolve back to its point of origin, it is necessary to evolve back the whole of its environment as well. Local information does not suffice. Similarly, unitary evolution in quantum mechanics holds only when nothing actually happens. If a measurement interaction happens, the system relaxes to some eigenstate of its evolution operator, say $a_{n}\Psi_{n}$. However, any state of the form $a_{n}\Psi_{n}+\sum_{k}a_{k}\Psi_{k}$ with $a_{n}^{*}a_{n}+\sum_{k}a_{k}^{*}a_{k}=1$ can collapse to $a_{n}\Psi_{n}$. Evolving backwards from $a_{n}\Psi_{n}$ leaves the system in eigenstate $a_{n}\Psi_{n}$. Time reversibility is far from universal.

Consider the following classical example. Imagine a J-shaped ramp with the long arm angled downwards towards a very large table. Over the bend in the ramp, suspend a spring laden trap door and over the door suspend a heavy ball. The ball is allowed to drop onto the door, pushing it open, falling to the ramp and rolling down onto the table, where it rolls away from the ramp. Meanwhile, the spring on the door has caused it to close. When the ball is some distance from the ramp, reverse all of the motions of the ball, ramp, spring, and door. At this time, only the ball is in ballistic motion. Everything else is in thermal motion. The direction of motion of the ball is reversed, but the thermal motion of everything else persists. The ball will roll up the ramp, up the inclined J portion and, if the length is correctly chosen, will fall off the other end. What it will not do is jump up off the ramp and pass through the suddenly open door. Thermal motion will not generate the force needed to open the door. The original motion is irreversible. To reverse it, *every* motion of every component must be reversed at every moment of time, but that requires knowledge of the entire history of motion of the door, ramp, and ball and must be imposed artificially. Note that entropy has nothing to do with this. The *global* arrangement of the door, ramp, and ball necessitates that this motion is irreversible, regardless of the local time reversibility of the equation of motion.

The theory of manifolds provides a cogent example of underdeterminism by local rules. Every manifold locally looks like $R^{n}$ for some fixed integer *n*. However, the global structure can come in an infinite variety of forms. Local time reversibility simply does not determine whether or not there is global time reversibility. The associated universe may be block or involve becoming; time reversibility does not serve as a constraint.

### 3.3. Arguments from Relativity

Arguments arising from the special theory of relativity are the most frequently cited as demonstrating the non-existence of time in physical reality. These arguments appear to fall broadly into three groups: arguments for the interchangeability of space and time, arguments against universal time, and arguments against simultaneity (and, thus, the now).

The interchangeability of space and time is most strongly argued from the perspective of general relativity, but the Lorentz equations provide an argument within special relativity. Recall that, if there are two observers, one considered at rest and the other in uniform motion along the x-axis relative to the first, then the co-ordinate values in the two frames of reference are related by
$$\begin{array}{c} x^{\prime }=\gamma \left(x-vt\right)\\ t^{\prime }=\gamma \left(t-vx/c^{2}\right)\\ y^{\prime }=y\\ z^{\prime }=z \end{array}$$
where $\gamma ={\left(\sqrt{1-(v^{2}/c^{2})}\right)}^{-1}$.

This is sometimes invoked to suggest that time and space are somehow interchangeable. However, space and time are not mixed up in the rest frames of either observer. They are still distinct, as is seen in their treatment in the invariant Minkowski metric, where time and space have opposite signs. The apparent intertwining is due to the need to take into account the finite speed of propagation of signals and the relative distance that such signals must travel. The Lorentz equations do show that the concept of an observable, consensual, absolute time is not tenable since the two observers will report two different time intervals for the same observed event. However, this does not preclude the possibility of an unobservable, absolute, global time ([7,41]) or of an absolute local time ([4]). The demand that an absolute time be observable runs afoul of the Fallacy of Misplaced Omniscience. It is only necessary that fundamental entities be influenced by this absolute time, but since human observers are emergent upon these fundamental entities, there is no reason to assume that phenomena at this lowest level might not be protected in the sense of Laughlin [42].

The idea of becoming has often been confused with a related idea, that of a flow of time. Newton wrote about the flow of time, but this has always been problematic since flow requires time, and by what time can the flow of time be judged? From this, confusion arises on the objectification of time, viewing time within the objectivist worldview and treating it as if it were an object with properties. Becoming is the statement that entities come into being, most often persist for a duration, and then fade away. In doing so, change occurs, and time marks the relation between these changes. The entities do something (change), but time itself does not *do* anything. Time is a relation. Recall the earlier discussion, which suggested that space is what distinguishes entities from one another, while time is what distinguishes the states of an entity itself. Time does not flow, but entities may still happen, and they may exist at the time that they happen but not at any prior or subsequent time.

Another argument used to support the concept of a block universe is based on the fact that two observers moving at high speed relative to one another may ascribe a different temporal order to events: an event might be in the past light cone of one observer but in the future light cone of another. Since past and future are thus relative notions, all events must co-exist. A basic assumption is that an event must exist in order for it to be observed. This is not a problem for events in the past light cone, which can clearly be said to exist prior to being observed. However, what of events in the future light cone? It is not possible to include an event in the future light cone unless it is observed (or unless one assumes a point of observation outside of the universe, again invoking the Fallacy of Misplaced Omniscience) which would imply that it must exist prior to being included in the future light cone. In order to assume the existence of both past and future light cones as real (and not as mathematical fictions), one must then first assume the existence of a block universe; again, begging the question.

However, to say that an event is in the future light cone of an observer merely says that a signal from said event is not observed by the observer until some time in its future. All one can say is that the event must have existed prior to the observation of the signal. One cannot state that the event itself lies in the future of the observer. If, at the time of observation, another observer declares that the event lies in its past light cone, then it has already been observed by that second observer, and therefore it must have existed prior to the time at which this information is exchanged between the two observers (which synchronises the null point of the two observers). That the event is not seen by the first observer until some time in its future is irrelevant. The event already exists. This is compatible with a block universe in which becoming occurs.

There is a deeper problem, however.

The Lorentz transformations are symmetrical, so the differences that appear to one observer in regard to measurements by the other are the same if the roles are reversed. The Lorenz contraction (or, more properly, rotation, according to Terrell [43]) is probably illusory [44]. Time dilation, however, appears to be physical, as observed in the slowing of half life in high-speed cosmic rays. The symmetry of time dilation leads to the famous twin paradox. Arthur [4] presents a very nice explanation of the twin paradox, pointing out that the histories of the two twins through space–time are quite different; their *lived experience* is different from one another, with one twin undergoing acceleration and deceleration and the other remaining inertial. The difference in observed proper times reflects this difference in lived experience. It reflects a difference in the *histories* of the two twins. If one accepts the fundamental criterion that what makes something real is that it makes a difference [26], then the difference in these paths is something real, not merely between curves in a space–time diagram. However, this suggests that something actually *happened* to these two twins in order to cause this difference. That could only occur in a universe with becoming, because in a block universe, nothing happens. If everything simply is, why should one path make any physical difference from any other path? Arthur writes [4] (p. 128), “It should not be thought, however, that it is the acceleration itself that produces the resulting dilation, as opposed to the difference in the length of the spacetime paths resulting from the fact that one twin has taken a path that is at least at some point non-inertial”. However, for a path to be non-inertial, it requires a cause, and the consequence of that cause will be acceleration or deceleration. The path is an effect, not a cause, but it expresses a difference in history, which is reflected in the difference in proper time. Time dilation is compatible with either a block universe or a universe with becoming, because it is not a feature of time itself, but rather of the history of an entity, and that history will unfold in either model of universe.

Yet, there is still another deeper problem, which is that the relativistic frames of reference, which are so widely referred to in making relativistic arguments, do not exist in reality. This has been well known for a long time [7,45], but it appears to have been ignored or glossed over in most relativistic arguments. Mathematically, as always, it is possible to conjecture or assume the existence of these frames of reference based around a single event, which serves as a null point and is then expanded into past and future light cones. These frames are analogous to the local charts used to define the topological structure of a manifold. As stated many times here, a mathematician is free to do as they choose when working with a mathematical construct. It is an entirely different matter to assert that these constructs correspond to something in reality. Whether the universe is static (block) or becoming, the existence of a fixed speed for signals places a fundamental limit on the kinds of frames of reference that may be available to any real observer. Notice that the argument that is used to show the non-existence of a consistent notion of simultaneity for space-like separated events also shows that there is no consistent notion of distance for those same events. It is not merely that, at a given point in time, one cannot know what is happening *now* because now has no meaning; one cannot, in fact, know anything that is happening at any space-like separated point anywhere and any when. We approximate in our day-to-day lives because the time differences are so small as to be inconsequential; but not always. Observable lags occur in broadcast, telephone, and internet signals. It is only the proper time that is invariant and thus meaningful [46].

However, the invariance of proper time is not enough. There can be no complete relativistic frames of reference in the real universe. The reason is simple. Suppose that an event lies within the mathematical future light cone of an observer. In order for the observer to have knowledge of this event, barring clairvoyance or the divine right of mathematicians, it is first necessary for some signal to propagate from the event to the observer. Unless one allows for the possibility of signals to propagate backwards in time (which I have argued above is implausible), then it will take at least the proper time between the observer and the future event for that signal to propagate to the observer (or, equivalently, for a signal to propagate from the observer to the event). In the absence of that signal, the observer cannot actually know that the event does or will exist at that space–time location. Thus, it is impossible for a real observer to actually construct their future light cone. To assert otherwise again runs afoul of the Fallacy of Misplaced Omniscience. In addition, while, in theory, every event in the past light cone of the observer could have sent a signal to the observer, making its existence at the recorded space–time location, it is highly likely that signals are deflected, destroyed, or never sent, so that the observer never receives information about said event. In that case, it cannot incorporate said event into its past light cone. To believe otherwise again runs afoul of the Fallacy of Misplaced Omniscience. One could argue that the observer could simply move forward in time, accumulating ever more information and building up its frame of reference, but in an infinite universe it will never complete this task because it will never run out of time and space, and in a finite universe it will run out of time before it runs out of space and still not complete the task. Thus, the very best that an observer can do is construct a partial past light cone at each point along its space–time trajectory. Any argument about the nature of time in our universe must take that into account. However, note that this is true even in a block universe (again, unless one allows for signals backwards in time), and so the absence of such frames of reference poses no implications as to whether the universe is static or becoming.

The mathematician is free to use frames of reference when talking about mathematical entities, such as Riemannian manifolds, causal spaces, and Minkowski spaces, but great care must be taken when translating that knowledge into statements about the nature of physical (as opposed to mathematical) reality. Ignoring this important point again invokes the fallacy of misplaced concreteness. This is a problem in cosmology, where one frequently reads about the current state of the universe. The simple truth is that we have absolutely *no* information about the current state of the universe beyond perhaps the local region of our solar system. Any statement otherwise is mere conjecture. The only thing we know is a slice through our local past light cone, a complicated projection of aspects of the past onto our present.

This important point was expressed by Winger decades ago [47]:

The basic premise of this theory is that coordinates are only auxiliary quantities which can be given arbitrary values for every event. Hence the measurement of position, that is, of the space coordinates, is certainly not a significant measurement if the postulates of the general theory are adopted: the coordinates can be given any value one wants. The same holds for momenta. Most of us have struggled with the problem of how, under these premises, the general theory of relativity can make meaningful statements and predictions at all. Evidently, the usual statements about future positions of particles, as specified by their coordinates, are not meaningful statements in general relativity. This is a point which cannot be emphasised strongly enough and is the basis of a much deeper dilemma than the more technical question of the Lorentz invariance of the quantum field equations. It pervades all the general theory, and to some degree we mislead both our students and ourselves when we calculate, for instance, the mercury perihelion motion ... Expressing our results in terms of the values of coordinates became a habit with us to such a degree that we adhere to this habit also in general relativity, where values of coordinates are not *per se* meaningful (p. 54) (SIC).

If there is any illusion, it is not in the experience of time; it is in the belief that there exist frames of reference that include present and/or future events and space-like separated events.

Relativity is frequently invoked to prove that a universal, absolute time does not exist. Nevertheless, several examples have been found of manifolds with a universal time structure that are compatible with the equations of general relativity. These include causal, dynamical triangulations, causal sets, and shape dynamics [3,48,49]. In spite of its great success, general relativity does have competitors that do not need to postulate an objectified space–time. These are MOND [50] and Taiji gravity [51], both of which consider modifications to the formula for acceleration itself rather than to time and space. Arthur points out that, in special relativity, time bifurcates into co-ordinate time and proper time [4]. Arthur argues that many of the problems in understanding relativity arise because of a failure to keep this bifurcation in mind, often confusing these two concepts. Co-ordinate time is relative and observer-dependent. Proper time, on the other hand, is invariant and universal. Proper time is the time of the process. We do not directly observe proper time because it is protected [42]. We can only measure the duration of time through the use of clocks, which are physical processes and that therefore only show us co-ordinate time. Clocks do not exist in some independent, external reality. They are physical processes within our universe, our reality, and subject to the same constraints that we are. Of course, if we are only interested in *our* proper time, then we can always measure it by means of a co-moving clock. What we cannot do, however, is consistently measure the proper time of an independent entity. Arthur has suggested a universe with multiple, local, and absolute times, and the process formulation provides exactly that.

The spatialisation of time reached its apotheosis in general relativity, whose fundamental equation, $R_{\mu \nu }-\frac{1}{2}Rg_{\mu \nu }-\Lambda g_{\mu \nu }=\kappa T_{\mu \nu }$, where $R_{\mu \nu }$ is the curvature tensor, $g_{\mu \nu }$ the metric tensor, and $T_{\mu \nu }$ the mass-energy tensor, describes a unified geometrical space–time. In general relativity, there is no becoming; space–time and the events within it simply *are*. The universe of general relativity is a block universe with a static time. General relativity per se cannot be used to prove the non-existence of time since it assumes this. However, the success of the predictions of general relativity has strengthened belief in the mathematical theory, and has also encouraged belief in the mathematical theory as an ontological theory, that is, believing that space–time *is* a four-manifold and the universe is a block. However, yet again, the fallacy of misplaced concreteness raises its head.

Note, though, that just as the relativistic frames of reference do not exist in the real universe, neither do the space–time metric nor the mass-energy tensor. These are mathematical constructs, mathematical models. Neither can ever be determined in reality, for the same reasons as given for the non-existence of relativistic frames of reference: a real observer can never acquire the information needed to form either of these entities in reality. We can make toy models and we can even pretend that these models represent our universe, but we can never obtain the information to determine these for our universe. Moreover, what makes a trajectory in space–time a trajectory of an entity is the continuity of passing from one space–time location to another, the ontological and epistemological coherence that correlates the various space–time locations into a whole, representing the history of an actual entity. This coherence lies internal to the entity; it is not imposed by fiat by some external observer. These events happen, and they happen in sequence from past to future for each entity. They do not happen in reverse order. It is this very happening that provides the coherence linking the components of a trajectory into a coherent whole. Denying this, one is left merely with a collection of points and nothing to distinguish one collection from any other collection. Simple physical characteristics are not enough. Think of the path of development of any complex organism. Mere appearance cannot link developmental forms, neither for most insects, most mammals, nor most reptiles. History is not a record of what *is*; it is a record of what *was*.

### 3.4. Quantum Mechanics Arguments

It is not possible to do justice to quantum mechanics within the limits of a short paper. Quantum mechanics has become a vast collection of disparate theories and models, loosely connected through the concept of quantisation of observables. The mathematical representation of quantum mechanics using Hilbert space has become dogma. Arguments against the existence of time based upon quantum mechanics face a fundamental problem, which is the lack of a consensual interpretation of these formalisms which link them to some sort of ontology. The standard dictum of “shut up and calculate” offers no assistance in this regard. If quantum mechanics is viewed merely as a formal tool for carrying out calculations of the probabilities of various observed quantities, then it cannot be invoked to make assertions about the nature of time, since calculations can be carried out in any logically consistent manner with any model of time that is convenient for the calculation. There is no consensus yet as to an ontological interpretation of quantum mechanics, but research over the past 40 years into foundational questions has shed light on what a possible ontology might include. At the very least, it is clear that most measurements are quantised. Measurements are, in general, non-commutative (though that has been known in the social sciences for at least a century) and contextual (whether what I have termed Type I contextuality, which refers to a dependence of a random variable representing the same measurement or observation of a system taking on different values depending upon the context in which such operations take place [52,53], or Type II contextuality, which relates specifically to violations of Bell-type inequalities and which also appears in classical settings such as decision making [54,55]), and there is some form of nonlocality, though there is no agreement as to what this non-locality is or how to interpret it ontologically.

There are many different formal representations of quantum mechanics: the standard Schrödinger equation, the operator (matrix) approach of Heisenberg, the phase space approach of Wigner [56], the diagrammatic approach of Coecke [57], and the channel approach of quantum information [58], to mention just a few.

The Schrödinger equation describes two conflicting modes of determination, as defined by Bunge [59]: deterministic in the unobserved evolution of the wave function Ψ, and stochastic under conditions of measurement, where the Born rule interprets $|\Psi^{*}\Psi |$ as a probability distribution function. Both forms of determination are compatible with the objectivist worldview, but the interpretation of the wave function as an ontological entity raises problems because a probability is not a natural kind. Whatever it is (and the philosophical literature on probability is far from any consensus on this point [60,61,62,63]), it describes an aspect of a collections of natural kinds, but is not a natural kind itself. Nevertheless, over time, this wave function has been forced into the objectivist worldview and objectified, leading to endless confusion. This appears to be an example of the fallacy of misplaced concreteness, again confusing the description and the thing described.

Another source of confusion arising from the objectivist worldview is that of statistical independence. In the objectivist worldview, separate objects possess not only separate, objective properties but also statistical independence, if they are not interacting with one another or with some third party. Kolmogorov stated that the problem of statistical independence was the central problem of probability theory, although he failed to elaborate upon this point in detail [60] (beyond pointing out that statistical independence requires that the product rule hold for all possible products of independent random variables, not merely pair-wise; a condition that is seldom tested in practice). Statistical independence is an assumption that lies at the heart of the derivation of the Bell inequalities and is rarely challenged [64]. A processist worldview (as expressed in process algebra) presents a much more nuanced approach to questions of statistical independence.

In non-relativistic quantum mechanics, time appears as an independent parameter, much as it does in classical mechanics. As for classical mechanics, to assert this ontologically requires the assumption of a block universe, which cannot then be used to prove that time does not exist. In classical mechanics, the equations can be reformulated as integral equations, such as
$$\Psi \left(t^{\prime },x^{\prime }\right)=\int_{t}^{t^{\prime }}\int_{X}K\left(x^{\prime },t^{\prime }|x,t\right)\Psi \left(t,x\right)dXdt$$
where $t,t^{\prime },x^{\prime }$ are fixed and the integrals are over all of space and over the time interval $\left[t,t^{\prime }\right]$. In quantum mechanics, one can reformulate in terms of Feynman path integrals. In both cases, these formulations are compatible with becoming in which an event comes into being over some duration $\left[t,t^{\prime }\right]$. As for classical mechanics, non-relativistic quantum mechanics does not make any claims concerning the non-existence of time.

There are formulations of quantum mechanics that involve the use of two times, such as Bar’s two-time physics [65] or stochastic quantisation [66]. This second time allows for some form of stochastic evolution over some small time interval, allowing the system to have an actual trajectory, although it is driven by a stochastic process. This is consistent with either a block universe or a universe with becoming. There is another use of two times, however, which occurs in the transactional interpretation of quantum mechanics of Cramer [67] and in the two-time formulation of Aharonov and Vaidman [68]. Here, the two times are a past and a future time in a single time dimension. In both of these formulations, there is an interaction or transaction between the wave function of a system at some time $t_{1}$ and the wave function of the same system at some time $t_{2}$, where $t_{1}<t_{2}$. In both models, the difference $t_{2}-t_{1}$ can be observed, and so, if some interaction of transaction between these two wave functions occurs, this is only possible if both wave functions co-exist, i.e., only if the universe has a block form. Any denial of time asserted by such a formulation again begs the question. However, these two models can be viewed simply as a form of retrodiction: selecting out in the future a subset of observations conditioned on some criterion. Such a view does not conflict with a universe with happening, although explaining the correlations that occur can be problematic within an objectivist worldview, especially if counterfactual definiteness or statistical independence of alternatives are demanded. However, in a universe of becoming, especially within a processist worldview where events are *generated*, one must distinguish between the generator and what is generated, and this form of retrodiction remains compatible with a universe with becoming.

Major problems arise when one attempts to interpret the wave function within an objectivist worldview, and I have argued that this stems from the use of the Hilbert space representation. A system exists in some state described by a wave function Ψ. This wave function can be decomposed into a sum of eigenstates of a self-adjoint operator, say the Hamiltonian *H*, so $\Psi =\sum_{n}a_{n}\Psi_{n}^{H}$. However, any self-adjoint operator may be chosen, say *A*, so that one can also write $\Psi =\sum_{n}a_{n}\Psi_{n}^{A}$. In the context of measuring energy via the Hamiltonian, objectification of the wave function leads to the suggestion that the system *is* in a state of superposition of eigenstates of the Hamiltonian. However, it is perfectly correct to also assert that the system *is* in a superposition of eigenstates of any self adjoint operator such as *A*. Objectification of the wave function leads one to interpret the Hilbert space sum as if it is actually a direct sum of functions, that is, $\Psi ={⊕}_{n}a_{n}\Psi_{n}^{H}$. However, this is not the case in the Hilbert space formulation. All one has is Ψ, in point of fact, and from the value of Ψ at any given location, one cannot determine the relative contributions to the value of Ψ from each of the eigenfunctions. That can only be found by applying the operator globally to the entire function, but in any real observation, one will have a value for the wave function only at a single point of observation. The Hilbert space formulation conflates ontology and epistemology, and this has led to endless confusion. The process algebra approach is more nuanced and distinguishes between ontology and epistemology, so it avoids this confusion. This is important for understanding issues related to locality, contextuality, and time. The process algebra approach can reproduce the results of non-relativistic quantum mechanics within a processist worldview, in which becoming is a fundamental feature of the universe [69,70,71]. This demonstrates that at least non-relativistic quantum mechanics is compatible with a universe with becoming.

Some researchers in quantum gravity have argued that time does not exist, or at least there is no need for time in order to carry out the physics [2]. However, there are many models of quantum gravity that are compatible with a universe with becoming [48,49]. Arguments for the non-existence of time often make reference to the Wheeler–DeWitt equation $\hat{H}\left(x\right)|\Psi >=0$, where |Ψ> is the wave function of the universe and $\hat{H}$ is the Hamiltonian constraint in general relativity. The Hamiltonian no longer provides a time evolution, so the Wheeler–DeWitt equation describes a timeless universe. Setting aside questions of what the wave function of the universe even means ontologically, to assert its existence requires some form of block universe in order for the values of the wave function to be actually defined at every space–time location. This is again a clear example of begging the question. Recourse to conjectures such as the multiverse does not help because those additional universes are all block universes.

I have considered many of the arguments that have been put forward in favour of the non-existence of time. In each case, I have suggested that the arguments fail to prove their case as they run afoul of the fallacy of misplaced concreteness, that of misplaced omniscience, or provide examples of begging the question. These arguments do not prove the non-existence of time, nor do they prove the existence of time. Physics appears agnostic on this question. The nature of time underlying any particular model or theory is a matter of choice, similar, to my mind, to the situation of the continuum hypothesis in mathematical logic. One can include it as an axiom or exclude it and still achieve a consistent theory. Likewise, one can assume a block universe or a universe with becoming; physics may make no definitive claim as to which assumption is “correct”. The belief in the non-existence of time stems, in large measure, from an excessive adherence to the objectivist worldview, the worldview of inanimate matter and of mathematical objects. However, there are other worldviews that might lead to greater understanding. One particular worldview that encompasses the biological side of existence is the processist worldview, to which I now turn.

## 4. The Processist Worldview

Table A1 provides a comparison of three different worldviews: the objectivist, the subjectivist, and the processist worldviews. The objectivist worldview has been extensively discussed in the previous sections. The subjectivist worldview is a worldview of human subjective experience. It is a worldview of qualia and interiority. If the appearance of time’s existence is illusory, then it will have some representation within the subjectivist worldview. However, to explore that would take us well beyond the scope of this article. The third worldview, the processist worldview, is the worldview of organisms, of living matter. Aspects of the processist worldview find their place in the philosophy of Heraclites, and the idea of becoming is fundamental to Therevada Buddhism [72]. Its strongest proponent in modern times was Alfred North Whitehead, who espoused these ideas in his influential work “Process Theory” [24]. In the years since Whitehead published his philosophy, there have been a number of physicists who have considered his ideas. Perhaps the earliest to express ideas that are at least sympathetic to those of Whitehead was Niels Bohr. There is much in common between Bohr’s notion of atomicity and Whitehead’s notion of the wholeness of actual occasions [73]. Indeed, in relation to biology, Bohr wrote, “With regard to the more profound biological problems, however, in which we are concerned with the freedom and power of adaptation of the organism in its reaction to external stimuli, we must expect to find that the recognition of relationships of a wider scope will require that the same conditions be taken into consideration that determine the limitations of the causal mode of description in the case of atomic phenomena” [74] (pp. 118–119). Bohr suggests that biology and quantum mechanics will share some fundamental characteristics, at least in respect to epistemological concerns. However, I agree with Rosen [75] that Bohr had the relationship reversed: it is biology that has much to teach quantum mechanics, not the other way around.

Abner Shimony showed an initial sympathy towards Whitehead [76] but later rejected his ideas. One reason being that he did not accept the possibility that particles such as electrons could be emergent upon a ground of entities such as actual occasions. David Bohm, too, in his notion of the implicate order, expressed many ideas that are also compatible with a processist worldview, at least insomuch as he considered the importance of the concept of becoming [77]. In recent years, a number of physicists and philosophers have taken up the idea that time is real and fundamental [78,79,80,81] or have explored models of process directly [69,70,81,82,83,84,85]. Some make explicit use of Whitehead’s process idea, while others simply argue for a general notion of process or of becoming.

Whitehead considered process to be the generator of reality. Unlike the objectivist worldview, which considers the entities of reality to simply exist, Whitehead considered becoming to be logically prior to being. In other words, the elements of reality do not simply exist; instead, they must come into existence through a process of becoming. Prior to becoming, they have no existence. Subsequent to becoming, they exist briefly, following which they again fade from existence. Reality is a continual succession of becoming, being, and fading away. The basic elements of reality that are generated by processes are termed actual occasions. These occasions have several characteristics, many of which are shared by organisms. They are:Actual occasions are both ontological and epistemological (informational) in character.Actual occasions are transient in nature. They arise, linger just long enough to pass their information on to the next generation of actual occasions, and then fade away.Process theory posits the existence of a transient now structured as a compound present: the current generation of actual occasions, the generating process, and the next generation of actual occasions.Actual occasions are holistic, discrete, and finite, possessing a “fuzzy” extensionality.Actual occasions are not directly observable. Only interactions among processes are discernible.Observable physical entities are emergent upon actual occasions.Information propagates causally from prior to nascent actual occasions as a discrete wave.Information from prior actual occasions is incorporated into nascent actual occasions through the act of prehension

The concept of prehension is the most difficult and subtle of Whitehead’s notions. Roughly speaking, there are two aspects: one that is concrete and one that is abstract. The concrete aspect is associated with the physicality or materiality of actual occasions or of the entities that supervene upon them. It is expressed in form and action. The abstract aspect is related to conceptualisation, to interpretation, to meaning, and to situating the content of the actual occasion within a coherent and consistent logical system. In the process algebra described below, these aspects are captured, in part, in terms of the intrinsic (concrete) and extrinsic (abstract) characteristics of informons.

Neither actual occasions nor processes are beables in the sense of Bell [86]. Bell writes

So it could be hoped that some increase in precision might be possible by concentration on the beables, which can be described in “classical terms” because they are there. The beables must include the settings of switches and knobs on experimental equipment, the currents in coils, and the readings of instruments.... The word “beable” will also be used here to carry another distinction, that familiar already in classical theory between “physical” and “non-physical” quantities. (SIC)

Bell’s view remains an objectivist view. Actual occasions are not physical entities in this sense. They do not possess properties. They do not possess dynamics, whether intrinsic or extrinsic. They do not possess motion. Nevertheless, they are elements of reality in the sense of [26] because they make a difference; they influence the subsequent actions of processes. Yet, they are primarily informational entities, serving more like a scratchpad for carrying out calculations; they exist long enough to fulfill this purpose and then are discarded. Processes as well are not beables in the sense of Bell. They are generators of events, but they are not events themselves. They generate a space–time structure; they do not exist within a space–time structure. This is consistent with a suggestion by Bancal et al. [87]. Again, following [26], they make a difference, particularly because they generate actual occasions and interact with other processes, altering them as a consequence. Thus, they are capable of making a difference. Actual occasions and processes are not objectivist entities, they are processist entities. Nevertheless, processes may be viewed as observables in a sense. They cannot, however, be Bell-type observables. Bell wrote [86]: “‘Observables’ must be made, somehow, out of beables” (SIC), but processes are formed from processes and so are not made of beables. Nevertheless, it is through the interaction between specialised processes called measurement devices and some other process called a system that measurement values are obtained. These measurement values are represented in the form of signs, which, in turn, are simply additional processes. So, processes observe, and processes are that which is observed.

The idea of process is best expressed in the behaviour of organisms such as that of a social insect colony, an example of collective intelligence. The ability of ant colonies to carry out complex forms of decision-making has been studied intensively for several decades. Ant colonies must build nests, care for broods, forage for food, and defend against attack. In choosing a new nest site, if conditions permit, they may make decisions that are consistent with ideas of rationality (and, in so doing, appear to act in a manner consistent with an objectivist worldview). However, when conditions are not suitable (such as pressure due to time or limited resources), decisions are more often made that are simply “good enough”. Such decision making is wholly dynamic: it occurs “on the fly” without recourse to implicit or explicit knowledge representation or planning. There is no central authority; the workers contributing to the decision are different and act differently on each and every occasion [52,88,89,90]. Collective intelligence is an embodiment of process; every decision is transient, generated, and contextual.

The process algebra was developed explicitly to capture the features of organisms noted above: generativity, transience, emergence, becoming, development, openness, contextuality, locality, and non-Kolmogorov probability. It draws on many sources: Whitehead’s process theory [24], Trofimova’s functional constructivism [91,92,93], combinatorial game theory [94], contextual probability theory [53], non-Kolmogorov probability [62], casual sets [49], and interpolation theory [95,96,97]. It has been applied to non-relativistic quantum mechanics [69,70] and is beginning to be applied to collective intelligence [52]. In process algebra, actual occasions do not correspond to “moments” or “instants” of time. They are not mathematical points but rather are fuzzy, extended entities with duration and spatial expansiveness. It can be shown that to each process there corresponds a local, invariant, causal structure that possesses a definite notion of simultaneity [98]. Thus, the temporal structure derived from process algebra will conform to a picture of local becoming as proposed by Arthur [4].

Table A2 compares characteristics that are typical of mathematical and classical objects, quantum systems, and biological systems. Note again the similarity between mathematical and classical objects, both of which are well captured by the objectivist worldview. The characteristics of both quantum systems and biological organisms stand in marked contrast to those of objects. Their characteristics are more typical of those expressed within the processist worldview. I suggest that an implicit bias towards the objectivist worldview has resulted in interpretations of quantum phenomena that bear little homology to the actual phenomenology, leading to many of the conundrums and conceptual confusions that have plagued quantum mechanics since its inception. I believe that a reframing of quantum mechanics within a processist worldview, which possesses a greater degree of homology with quantum phenomena, will lead to conceptually coherent interpretations and eliminate paradoxes and other quasi-mystical interpretations.

One advantage of the processist worldview is that it puts reality back into physics. It is not the reality of the objectivist worldview, however. Processes are generators, not objects. Actual occasions are tokens, not objects. Process propagate information locally between actual occasions locally, and actual occasions provide counterfactual definiteness, but only from moment to moment and not in any enduring sense. Processes are contextual, and what is generated is contingent. The processist worldview restores time to its rightful place at the foundation of reality. It does so with a local, but not necessarily global, absolute time, which is formed not of mathematical instants but rather of durations. It gives a central role to transience and to transients as the proper entities for study. It posits a universe in which becoming is logically prior to being. It shows that the biological realm had it right all along.

## 5. Conclusions

Several prominent arguments against the existence of time have been examined from the perspective of the worldviews, implicit or explicit, underlying them. These arguments arise within the philosophy of science and physics, particularly classical, relativistic, and quantum. In each case, it is shown that there are implicit assumptions fundamental to the arguments arising from their underlying worldview that run afoul of either the Fallacy of Misplace Omniscience, the fallacy of misplaced concreteness, or serve as examples of begging the question. These weaken or nullify the arguments and their conclusions. In the end, all one can truly state is that physics, in its current state, neither denies nor affirms the existence of time. In the absence of direct empirical evidence, the best that we can do is treat the assumption of the existence of time as an independent axiom, similar, perhaps, to the place of the continuum hypothesis in mathematical logic. The discussion of process algebra shows that a physics in which time exists and becoming is fundamental is possible, and thus the non-existence of time is most certainly not a pre-requisite for doing physics.

## Data Availability

This study generated no data.

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
