# Peer review of "Process and Time"

_entropy, 2023, doi:10.3390/e25050803_

Round 1

Reviewer 1 Report

See attached pdf.

Reviewer 2 Report

The topic of the paper is interesting.

But the paper is too long it is rather a very extended discussion on different opinions than a scientific paper.

A considerable effort of synthesis must be made (now the paper is 40 pages  long).

For example ther could be several tables comparing the different hypothesis and models used for defining the role of time with the proposed interpretation.

English seem OK.

Round 2

Reviewer 1 Report

The author  has significantly revised the first version of the manuscript, fixed all the issues the reviewer  had find in the first version.  The current version is significantly shortened,  list of references  improved, vague places of the first version rewritten  and clarified.  The reviewer  concludes that the current version of the manuscript deserves publication in Entropy. 

Reviewer 2 Report

A big effeort has been made to make the paper more readable.